# Controlling a μRTS agent using Decision Transformers

## Abstract

Decision Transformers (DT) are a Return-Conditioned Supervised Learning (RCSL) technique. A DT policy predicts actions by attending to a limited history of tokens that encodes states, actions and returns-to-go. Two existing extensions of DT, namely Online Decision Transformers (ODT), and Critic Guided Decision Transformers (CGDT), are re-implemented and applied to the Gym-μRTS environment. In CGDT, a critic learns to predict reward distributions conditioned on sequences of interwoven states and actions to overcome DT's issues with stochasticity. In ODT, an additional entropy term and hindsight reward relabeling enable online fine-tuning. A dataset is generated from games between CoacAI and Mayari, two previous μRTS competition winners, on procedurally generated 8x8 maps. We further explore the combination of both CGDT and ODT methods to create an Online Critic-Guided Decision Transformer (OCGDT). Training proceeds in three phases: (1) supervised learning of the critic using a fixed dataset of 3,000 trajectories, (2) supervised learning of the policy from the same dataset, and (3) online fine-tuning of the policy, using the dataset as a starting point for a replay buffer. The critic and offline networks are validated against 500 held-out trajectories, while the final policy's performance is measured by its win rate against four benchmark rule-based bots (CoacAI, Mayari, lightRushAI, and workerRushAI). The agent obtains a win rate of $26.2\% \pm 4.3\%$ against CoacAI and a win rate of $40.1\% \pm 4.8\%$ against Mayari over 4 seeds per match-up and 100 games per seed for a total of 400 games per match-up on held-out procedurally generated 8x8 maps. This matches the performance of Implicit Q-Learning (IQL). The agent also obtains a win-rate of $51.6\% \pm 4.9\%$ when matched up directly against IQL.

## 1 Introduction

Real-time Strategy (RTS) games feature large action spaces, partial observability, and require both short- and long-term decision-making (Ontañón et al., 2018; Huang et al., 2021; Vinyals et al., 2019). This posits a strong challenge for reinforcement learning (RL) agents Vinyals et al. (2019). Traditionally, RL relies on online interactions with the environment to gradually improve a behavior policy based on the feedback returned by the environment. An agent's performance is measured based on this feedback. However, this feedback can be sparse, with long periods of exploration until a reward is finally encountered. This can be mitigated by training from offline data, which is preferred when online interactions are costly or risky (Levine et al., 2020).

### 1.1 Research Motivation

Our work focuses on handling sparse rewards over long horizons in Gym-μRTS. Online RL needs extensive bootstrapping for long-term credit assignment (Nair et al., 2020), while offline RL, though data-hungry, circumvents this issue (Levine et al., 2020). Decision Transformers (DT) perform better than offline RL in low-data regimes with low-quality demonstrations and handle long-term dependencies well (Bhargava et al., 2024). Sparse rewards can also be alleviated by adding synthetic ones—a technique known as reward shaping (Ng et al., 1999). Effective shaping demands domain expertise and risks biasing the agent, while a clear win/lose/draw signal sets unambiguous goals. There are approaches to mitigate this bias. Potential-based shaping methods are equivalent to Q-value initialization (Wiewiora, 2003; Ng et al., 1999), and Hu et al. (2020) adaptively refine rewards

through a bi-level optimization procedure. Rather than shaping an extrinsic reward directly, an agent can also generate an intrinsic reward to explore the state space more effectively through a metric such as surprise, novelty, or skill learning (Pathak et al., 2017; Burda et al., 2018b;a; Aubret et al., 2023).

Chen et al. (2021) claim that DT stitches together sub-trajectories in a minority of cases when finding the shortest path in a graph. This stitching behaviour does not hold true when there is an abundance of sub-optimal data in the dataset (Wang et al., 2024; Paster et al., 2022). In addition, uncertainty and approximation errors within the behavior policy introduce a form of stochasticity that resembles environmental stochasticity. Critic-Guided Decision Transformers (CGDT) (Wang et al., 2024) tackle this by pre-training a critic on an offline dataset to output a reward distribution that guides policy updates via an asymmetric expectile loss. Online Decision Transformers (ODT) (Zheng et al., 2022) modify the DT architecture by adding online fine-tuning, entropy regularization for exploration, hindsight relabeling, and stochastic outputs. We set out to explore the effectiveness of DT-based methods in Gym-µRTS and whether these two methods can be combined to produce a stronger model.

## 1.2 CONTRIBUTIONS

CGDT, ODT, and IQL are reproduced and adapted to the Gym-µRTS environment. The critic component of CGDT are combined with ODT's fine-tune-enabling components to create the Online Critic-Guided Decision Transformer (OCGDT). OCGDT is a novel method directed at tackling the short-comings of DT while still allowing for an online fine-tuning stage. OCGDT, CGDT, ODT, and IQL are evaluated against four benchmark rule-based bots provided with the Gym-µRTS package (CoacAI, Mayari, WorkerRushAI, and LightRushAI). The agents are also evaluated directly against each other. Additional to the four implementations of the AI agents is a Gym-µRTS dataset containing 3,000 trajectories of CoacAI and Mayari sampling maps from a set of 1,000 procedurally generated 8x8 scenarios. OCGDT matches the performance of IQL in half the wall-clock hours.

## 2 BACKGROUND

This section reviews the foundational concepts of our work. We begin with an overview of the µRTS and Gym-µRTS environments, justifying the choice of environment and describing the state and action space. Then, preliminaries are introduced for Return-Conditioned Supervised Learning (RCSL), Critic-Guided Decision Transformers (CGDT), and Online Decision Transformers (ODT), with the extensions addressing short-comings in standard Decision Transformers (DT). The mathematical notation used here differs slightly from the original papers to keep definitions consistent between both methods.

### 2.1 µRTS & GYM-µRTS

Agents like AlphaStar have demonstrated superhuman capabilities in StarCraft II. However, such an agent relies on vast computational resources (Vinyals et al., 2019) which are often unavailable to academics. µRTS is a minimalist, open-source RTS engine that reduces resource overhead and lowers the barrier of entry into RTS AI research (Ontañón et al., 2018; Huang et al., 2021). While a highly detailed simulation represents a challenge in itself, it is also a computational hurdle for conventional hardware. µRTS retains the core elements of the RTS genre while still being a computationally feasible environment. It is used in the IEEE Conference on Games (CoG) competition which, until the winner of the 2024 edition (Goodfriend, 2024), was dominated by rule-based bots.

Gym-µRTS (Huang et al., 2021) is a python interface for the µRTS (Ontañón et al., 2018) simulator, providing a fixed low-level action space where each unit must be controlled with primitive commands, decomposing the huge action space into smaller components. Invalid action masking prevents invalid actions, reducing the effective action space. Gym-µRTS expects a multi-discrete action set at every step. The policy outputs an action for each cell in the grid (Han et al., 2019). This approach is used by the authors of the Gym-µRTS environment to produce a benchmark RL agent (Huang et al., 2021).

The Gym-µRTS state space consists of 29 feature planes representing hit points (5), resources (5), unit ownership (3), unit type (8), current action (6), and terrain type (2) in every grid cell. The

action space is represented by a vector whose length depends on the range of the *ranged* unit. With a default range $a_r = 7$, the action vector has $29 + a_r^2 = 78$ feature planes representing the action type (6), move direction (4), harvesting direction (4), return resource direction (4), production spawn direction (4), production type (7), and relative attack position ($a_r^2$) in every grid cell.

## 2.2 RETURN-CONDITIONED SUPERVISED LEARNING

In offline RL, a behaviour policy $\pi_\beta$ samples transitions from an existing dataset $\mathcal{D}$ containing trajectories $\tau = (s_0, a_0, r_0, ..., s_t, a_t, r_t, ..., s_T, a_T, r_T)$ where $s_t \in \mathcal{S}$, $a_t \in \mathcal{A}$, and $r_t \in \mathcal{R}$ are the states, actions, and rewards returned at time-step $t$, and $T$ is the finite horizon of a Markov Decision Process. The goal of RL is to maximise the expected cumulative return $\mathbb{E}[R(\tau)]$, where $R(\tau) = \sum_{t=0}^{T} r_t$ and represents the cumulative return of the trajectory $\tau$ (Levine et al., 2020).

Return-conditioned supervised learning (RCSL) reinterprets the offline RL setup as a sequence-modeling problem (Chen et al., 2021). Rewards are replaced with returns-to-go $\hat{R}_t = \sum_{t'=t}^{T} r_{t'}$. During training, the return-to-go represents the rewards yet to be collected in a trajectory. During inference, it denotes the target reward to be obtained by a generated action sequence (Chen et al., 2021). This changes the trajectory representation to the following:

$$\tau = \left( \hat{R}_0, s_0, a_0, \ldots, \hat{R}_t, s_t, a_t, \ldots, \hat{R}_T, s_T, a_T \right) \tag{1}$$

The goal is now to minimize some distance measure between the policy distribution and the sampled trajectory of length $K$, where $K$ is a hyperparameter referred to as the context length (Zheng et al., 2022). The standard DT formulation is presented with both $l^2$ loss (Chen et al., 2021; Zheng et al., 2022; Wang et al., 2024) and negative log-likelihood (NLL) (Zheng et al., 2022; Wang et al., 2024). Online Decision Transformers use a scaled NLL loss, as below:

$$\mathcal{L}^{RCSL}(\theta) = \frac{1}{K} \mathbb{E}_{\tau_K \sim \mathcal{D}} \left[ -\sum_{k=1}^{K} \log \pi_\theta \left( a_k | \tau_{-K,k} \right) \right] \tag{2}$$

where $\pi_\theta$ is the learning policy parameterized by $\theta$. $a_k$ is the action sampled from $\pi_\theta$ at time $k$. $\tau_K$ is a sub-trajectory of length $K$. $\tau_{-K,t} := \tau_{max(1,t-K+1):t}$ is the latest $K$ time-steps until time $t$ containing the past $K$ tokens for each $\hat{R}_t$, $s_t$, and $a_t$.

The Decision Transformer (DT) is one such RCSL method, bypassing traditional RL's reliance on value function approximation or policy gradients (Chen et al., 2021). While DT has performed well in specific offline RL tasks, its reliance on per-trajectory returns-to-go limits its effectiveness in stochastic environments, as a high return may be due to a lucky sequence of transitions, skewing the policy towards unlikely outcomes (Paster et al., 2022; Brandfonbrener et al., 2022). In addition, DT struggles to stitch suboptimal trajectories. In deterministic settings, its success hinges on having optimal trajectories in the training set. Consequently, the learned policy may mimic environmental stochasticity because of uncertainty and approximation errors. These issues underscore the importance of integrating probabilistic methods to better capture expected returns (Paster et al., 2022; Brandfonbrener et al., 2022).

## 2.3 CRITIC-GUIDED DECISION TRANSFORMER

The Critic-Guided Decision Transformer (CGDT) addresses stochasticity by using a critic to estimate the return-to-go distribution for a trajectory conditioned on sequences of state-action pairs. CGDT bridges the gap between deterministic trajectory modeling and probabilistic RL strategies (Wang et al., 2024) with an approach that has shown superior performance in stochastic settings (Paster et al., 2022; Brandfonbrener et al., 2022).

First recall that by Bayes' rule, $p(a_t|\hat{R}_t, s_t) \propto p(a_t|s_t)p(\hat{R}_t|s_t, a_t)$. (Wang et al., 2024), suggest modeling the unknown distribution $p(\hat{R}_t|s_t, a_t)$ as a Gaussian distribution with learnable mean and variance $(\mu, \sigma)$ sampled from a critic $Q_\phi(\hat{R}_t|\tau_{-K,t})$. The critic is trained using an offline dataset $\mathcal{D}$

with NLL loss as the objective:

$$\mathcal{L}_Q^{CGDT}(\phi) = -\log Q_\phi(\hat{R}_t \mid \tau_{-K,t}) \tag{3}$$

The NLL objective is revised as an asymmetric loss that biases the critic towards fitting optimal or suboptimal trajectories depending on the composition of the dataset $\mathcal{D}$:

$$\mathcal{L}_Q^{CGDT}(\phi) = -\big|\tau_c - \mathbb{I}(u > 0)\big|\log Q_\phi(\hat{R}_t \mid \tau_{-K,t}) \tag{4}$$

where $\mathbb{I}(\cdot)$ is the indicator function (1 if the condition is true, 0 otherwise), and $u = (\hat{R}_t - \mu_t)/\sigma_t$. The asymmetry coefficient $\tau_c \in [0,1]$ penalizes over- or underestimates by scaling the log-likelihood. If $\tau_c < 0.5$, the model is penalized more heavily for over-estimating the returns-to-go (i.e. if $u < 0$) and vice-versa.

The critic guides the policy with the following objective:

$$\mathcal{L}_{\tau_p}^{CGDT}(u) = \big|\tau_p - \mathbb{I}(u < 0)\big|u^2 \tag{5}$$

Where $\tau_p$ is the expectile asymmetry parameter that biases the critic's estimation towards over- or under-estimating the true returns-to-go. The vanilla RCSL learning objective is used as a constraint to keep the resulting policy close to the data distribution:

$$\mathcal{L}_\pi^{CGDT}(\theta; \alpha) = \mathcal{L}^{RCSL}(\theta) + \alpha \cdot \mathcal{L}_{\tau_p}^{CGDT}(u) \tag{6}$$

where $\alpha$ is a coefficient controlling the critic guidance contribution.

## 2.4 ONLINE DECISION TRANSFORMER

Zheng et al. (2022) note that standard DT does not take into account online exploration. Online Decision Transformers (ODT) address this limitation in DT architectures with a number of additions. ODT extends the DT architecture by introducing stochastic policies, an entropy regularization term for exploration, and hindsight return labeling, where once a new trajectory is collected, the conditioned return-to-go is replaced by the actual return-to-go encountered in the trajectory. Any newly gathered experiences are stored as entire trajectories rather than individual transitions. When including online fine-tuning, ODT sees significant gains (Zheng et al., 2022) relative to Implicit Q-Learning (IQL).

Entropy regularization is calculated as the Shannon entropy of the distribution of the policy $\pi_\theta$. In addition, Zheng et al. (2022) scale both the policy objective and the entropy by $\frac{1}{K}$ to allow for easy comparisons to Soft Actor-Critic (SAC) (Haarnoja et al., 2018):

$$H_\theta^{\mathcal{T}}(\mathbf{a}|\tau_K) = \frac{1}{K}\mathbb{E}_{\tau_K \sim \mathcal{T}}\left[\sum_{k=1}^{K} H\left(\pi_\theta\left(a_k|\tau_{-K,k}\right)\right)\right] \tag{7}$$

Adding this entropy as a constraint to equation 2 produces the following two optimization problems:

$$\min_\theta \mathcal{L}^{RCSL}(\theta) - \lambda \cdot H_\theta^{\mathcal{T}}(\mathbf{a}|\tau_K) \tag{8}$$

$$\min_{\lambda \geq 0} \lambda(H_\theta^{\mathcal{T}}(\mathbf{a}|\tau_K) - \beta) \tag{9}$$

where $\beta$ is the entropy lower bound which ODT sets at $-\dim(\mathcal{A})$, where $\dim(\mathcal{A})$ is the size of the action set, and $\lambda$ serves as a temperature parameter. The ODT policy objective is defined as follows:

$$\mathcal{L}^{ODT}(\theta; \lambda) = \mathcal{L}^{RCSL}(\theta) - \lambda \cdot H_\theta^{\mathcal{T}}(\mathbf{a}|\tau_K) \tag{10}$$

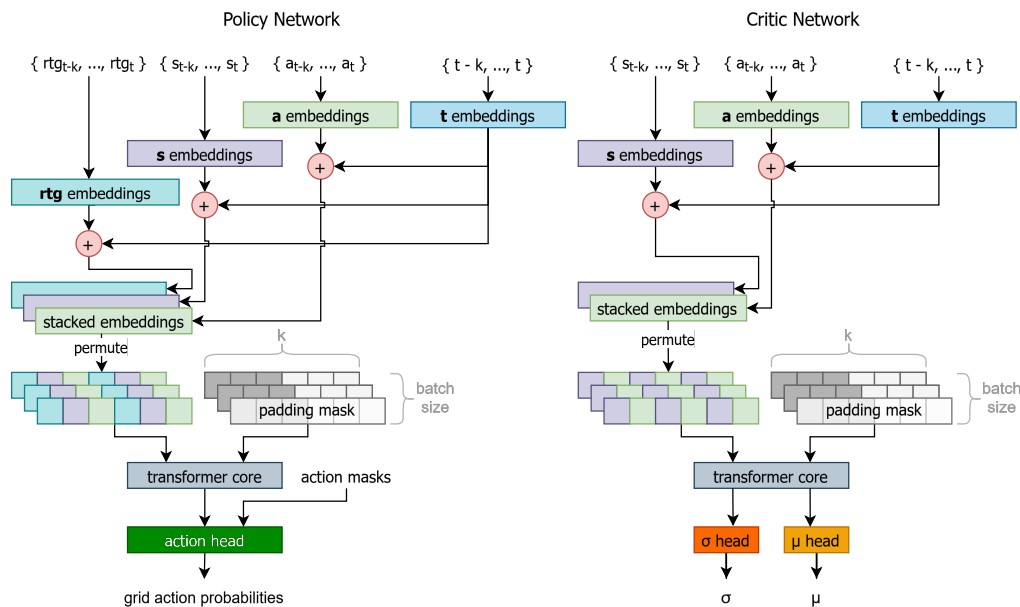

Figure 1: The critic network (right) makes use of the action and state embedding components and adds their output to a linear time-step embedding. The results are stacked and permuted to form a state-action sequence. The transformer core consists of a GPT2 transformer with four heads and two layers. The transformer core output is passed through a fully connected layer, a GELU non-linearity, and a final fully connected layer with two output heads. The policy network (left) is similar to the critic network, adding a linear return-to-go embedding. The stacked embeddings are permuted into a sequence of returns-to-go, states, and actions. Action masks are applied to the final action head, which outputs likelihoods for an action on each cell on the grid. The action head is composed of a series of fully connected layers with GELU activation functions, each doubling in output size, starting from 512 up to 2048. The final layer's output size is determined by the map dimensions and the number of channels per action vector.

## 3 ARCHITECTURE

This section first derives the equation obtained from combining ODT and CGDT. Then, the OCGDT and IQL architectures are explained.

### 3.1 OCGDT PRELIMINARIES

The loss functions of both ODT and CGDT can be combined to produce an agent that makes use of both online fine-tuning and a pre-trained critic. OCGDT uses NLL in its policy objective due to the multi-discrete action space. ODT and CGDT both use Equation 2 to constrain the actions close to the training distribution. The critic and entropy constraints are added separately to the loss, resulting in the following policy objective:

$$\mathcal{L}_\pi(\theta; \alpha; \lambda) = \mathcal{L}^{RCSL}(\theta) + \alpha \mathcal{L}^{CGDT}_{\tau_p}(u) - \lambda H^{\mathcal{T}}_\theta(\mathbf{a}|\tau_k) \tag{11}$$

ODT calculates the lower entropy bound to be $-\dim(\mathcal{A})$. Since Gym-μRTS has invalid action masking, this lower bound will change depending on the state, as the number of possible actions changes every step. In the reconstructions of ODT and CGDT, and in OCGDT, $\mathcal{A}$ is recalculated at every step to be the average number of valid actions over the last $K$ steps.

## 3.2 OCGDT Architecture

The agent consists of two models: The critic model and the policy model. The policy model selects an action conditioned on the past $K$ tokens of returns-to-go, states, and actions for a total of $3K$ tokens, as per DT (Chen et al., 2021). The first action token is a no-op. The critic predicts an expected reward which is represented as a Gaussian distribution. The critic has two heads: the standard deviation $\sigma$, and the mean $\mu$ of the Gaussian distribution. The distribution of rewards is conditioned on a sequence of $K$ state-action pairs, where the action is either sampled from a training set or predicted by the policy network, as per CGDT (Wang et al., 2024). Time-steps are linearly embedded and added separately to the returns-to-go, states, and actions to encode temporal information about the token sequence. The critic model and the policy model have separate parameters for their state representations. Policy logits are scaled by a temperature coefficient before passing through a softmax. During training, this is set to $1$, while during evaluation, it is set to $0.25$ for more decisive actions. Figure 1 depicts the policy and critic networks. Further details for the embeddings and heads are provided in the Appendix in Figures 3, 4, and 5.

Gym-µRTS returns an action mask for each state. Policy outputs are masked by subtracting a negative value with large magnitude from invalid logits. Action masks are stored alongside each transition of a trajectory to allow masking policy outputs during training as well.

## 3.3 IQL Architecture

Implicit Q-Learning (IQL) is an offline RL method that avoids querying $Q(s, a)$ for out-of-distribution actions. The policy improvement step is implicitly approximated by treating the TD target $r + \gamma V(s')$ as a random variable induced by the dataset actions and dynamics, and learning the value $V(s)$ as a state-conditional upper expectile of this distribution, making it focus on the high-return tail of the in-distribution actions. A Q-function is trained by regressing $Q(s, a)$ onto the same TD target. The policy is then improved via advantage-weighted behavioral cloning, using advantages $Q(s, a) - V(s)$ computed only on dataset actions Kostrikov et al. (2022).

The actor (policy network), critic (Q-network), and value functions re-use the same state- and action- (for the critic) embedding architecture as in OCGDT (see Figure 3 in the Appendix). The networks do not share parameters for state representation. The policy network (see Figure 6) consists of four fully connected layers layers with ReLU activations. The first layer has 256 hidden units, each layer doubling the hidden units. Its head outputs a likelihood distribution over actions, with a mask applied before the softmax function to prevent invalid actions. Twin Q-networks are utilized to reduce state-action value over-estimation (Hasselt, 2010; van Hasselt et al., 2016), taking the minimum of their predictions. The Q-network (see Figure 7) is updated with a soft update, interpolating towards the current Q-network parameters with $\tau = 0.005$. The Q-network takes the concatenated spatial embeddings of the state and action as input, followed by five fully connected layers: The first three have 512 units, and the last two have 256 and 1 unit respectively, with the final scalar representing the Q-value. ReLU functions are applied after every hidden layer. The value network (see Figure 8) mirrors the critic architecture with the action embedding excluded, given that it only operates on the state embedding.

## 4 Training

This section describes the process of dataset synthesis and provides an overview of the training set-up used to obtain the results for DT-based models and the IQL benchmarks.

## 4.1 Dataset Synthesis

A training set is synthesised for offline training. It contains 3,000 trajectories collected from games between CoacAI[1] and Mayari[2], two previous µRTS AI competition winners. Each game is played on a randomly selected map from 1,000 procedurally generated maps. The maps are of size 8x8. Starting conditions are restricted to one base and one worker per player and 4 resource nodes. The

---

[1]https://github.com/Coac/coac-ai-microrts
[2]https://github.com/barvazkrav/mayariBot

remaining cells in the grid have a small chance to be a wall. An additional 500 trajectories are held out and used as a validation set. The maps used by the trajectories in the validation set are sampled from a separate list of 1,000 procedurally generated maps. Another two sets of 1,000 procedurally generated maps are used separately for online training and evaluation respectively.

## 4.2 DT TRAINING

The critic is trained on a fixed dataset of 3,000 trajectories. A validation set of 500 trajectories is used for hyper-parameter optimization. A critic training run is 3,000 training steps. Using a Weights & Biases[3] sweep, Bayesian hyperparameter optimization is used to search among 4 hyper-parameters over 50 runs. Dropout is sampled from values between 0.1 to 0.3 with increments of 0.1. Weight decay is sampled uniformly between $1e^{-5}$ and $1e^{-3}$. The learning rate is sampled uniformly between $1e^{-6}$ and $1e^{-4}$. $\tau_c$ is sampled from values between 0.1 and 0.9 with increments of 0.1.

The best performing critic model is used to train the policy model. Offline training and validation of the policy is performed on the same training and validation sets as the critic. The policy's perfor-mance throughout training is measured against benchmark bots on a separate procedurally generated map pool. Each offline run is 5,000 training steps. Hyper-parameters are optimized in the same man-ner as the critic hyper-parameters. $\tau_p$ samples from values between 0.1 and 0.9 with increments of 0.1. The final hyper-parameter list is provided in the Appendix in Tables 3 and 4.

The training set is treated as a replay buffer during fine-tuning. A new trajectory replaces a trajectory in the replay buffer with every online episode. 4 online rollouts are performed every 50 steps. The online training run is 5,000 steps. The policy's performance throughout the run is measured in the same way as during the offline portion, by playing against benchmark bots on a separate map pool.

Agents are finally evaluated by playing 400 games, one seed for every 100 games, against the bots on another set of procedurally generated maps. Wins contribute 1 point, draws contribute 0.5 points, and losses contribute 0 points. Training is performed on a Windows 10 machine with an NVIDIA GeForce RTX 4090 GPU, 128GB of DDR4 RAM and an Intel Core i7-5820K CPU. A full training run, including all three phases and validation, takes 4.25 wall-clock hours with $K = 100$ and a batch size of 32. In total, there are 13,000 gradient update steps.

## 4.3 IQL TRAINING

To ensure a fair comparison between IQL and OCGDT, three variants of IQL are trained. The first variant, IQL 800k, is trained over an equivalent number of offline samples as the DT methods. With $K = 100$, a batch size of 32, 3,000 critic steps and 5,000 offline training steps, OCGDT sees 25,600,000[4] samples. With the same batch size, IQL needs to perform 800,000 steps. This takes 9 wall-clock hours. Hyper-parameter optimization is performed on this variant in the same manner as for OCGDT, over 50 runs. The dropout is sampled from values between 0.1 to 0.3 with increments of 0.1. The actor, Q-function and value function learning rates are all separately sampled uniformly between $1e^{-5}$ and $1e^{-3}$. IQL's $\beta$ parameter is sampled from the values between 1 and 5 with increments of 1. IQL's $\tau$ parameter is sampled from values between 0.1 and 0.9, with increments of 0.1. The hyper-parameters obtained here are re-used for the other variants. The second variant, IQL 400k, is trained for the same wall-clock duration as OCGDT, which is rounded to 400,000 steps. The final variant, IQL 13k, is trained for the same number of gradient updates as OCGDT. The final hyper-parameter list is provided in the Appendix in Table 5. A comparison of parameter counts is provided in Table 6.

## 5 RESULTS

This section discusses the results and implications of the ablative models and explores some learned behaviours. The main models of interest are CGDT, ODT, and OCGDT. Ablations are performed on components of OCGDT to understand their impact. IQL is used as a baseline state-of-the-art algorithm to compare results. Table 1 shows win rates against benchmark rule-based bots. CoacAI

---

[3]https://wandb.ai/
[4]$100 \times 32 \times (3,000 + 5,000) = 25,600,000$

Table 1: Agent win rates (%) against benchmark AI bots obtained over 4 seeds, 100 games each, for a total of 400 games on randomly sampled 8x8 maps. OCGDT *A* is an online-only agent, with a buffer large enough to hold the latest online trajectories. OCGDT *B* underwent double the fine-tuning steps. OCGDT *C* underwent double the fine-tuning steps but with an extended buffer size that prevented offline data from being replaced by online data. OCGDT *D* did not undergo fine-tuning. OCGDT *E* underwent double the offline training steps (10,000 instead of 5,000). OCGDT *F* is trained with $K = 20$. OCGDT *G* is trained with $K = 20$ for twice the duration. Interval represents 95% Wilson score interval. Best performance in bold.

| Method | CoacAI % | Mayari % | K | Buffer | Offline Steps | Online Steps |
|---|---|---|---|---|---|---|
| CGDT | $22.3 \pm 4.1$ | $40.8 \pm 4.8$ | 100 | 3,000 | 5,000 | 0 |
| ODT | $25.5 \pm 4.2$ | $\mathbf{46.3 \pm 4.9}$ | 100 | 3,000 | 5,000 | 5,000 |
| OCGDT (Ours) | $\mathbf{26.2 \pm 4.3}$ | $40.1 \pm 4.8$ | 100 | 3,000 | 5,000 | 5,000 |
| *OCGDT Ablations* | | | | | | |
| A (Online Only) | $3.2 \pm 1.7$ | $4.9 \pm 2.1$ | 100 | 4 | 0 | 5,000 |
| B (Double Online) | $15.3 \pm 3.5$ | $29.9 \pm 4.5$ | 100 | 3,000 | 5,000 | 10,000 |
| C (B + Larger Buffer) | $20.0 \pm 3.9$ | $40.8 \pm 4.8$ | 100 | 3,800 | 5,000 | 10,000 |
| D (No Fine-tuning) | $23.0 \pm 4.1$ | $43.3 \pm 4.8$ | 100 | 3,000 | 5,000 | 0 |
| E (Double Offline) | $16.7 \pm 3.6$ | $35.0 \pm 4.7$ | 100 | 3,000 | 10,000 | 5,000 |
| F (Short Context) | $22.3 \pm 4.1$ | $42.4 \pm 4.8$ | 20 | 3,000 | 5,000 | 5,000 |
| G (F + Double Steps) | $22.6 \pm 4.1$ | $41.1 \pm 4.8$ | 20 | 3,000 | 10,000 | 10,000 |
| *IQL Benchmarks* | | | | | | |
| IQL 800k | $21.5 \pm 4.0$ | $42.6 \pm 4.8$ | N/A | 3,000 | 800,000 | 0 |
| IQL 400k | $20.8 \pm 4.0$ | $35.9 \pm 4.6$ | N/A | 3,000 | 400,000 | 0 |
| IQL 13k | $7.9 \pm 2.6$ | $22.1 \pm 4.0$ | N/A | 3,000 | 13,000 | 0 |

Table 2: Cross-agent win rates (%). Player 1 in row label and Player 2 in column header. A game is counted as a loss for both players if they kill each other's last unit simultaneously, thus resulting in win rates that do not add up to 100%. Win rates are obtained over 400 games on randomly sampled 8x8 maps. Interval represents 95% Wilson score interval. Best performance in bold.

| vs % | CGDT | ODT | OCGDT | IQL 800k | IQL 400k | IQL 13k |
|---|---|---|---|---|---|---|
| CGDT | / | $47.0 \pm 4.9$ | $46.4 \pm 4.9$ | $50.5 \pm 4.9$ | $49.1 \pm 4.9$ | $60.7 \pm 4.8$ |
| ODT | $50.7 \pm 4.9$ | / | $47.5 \pm 4.9$ | $49.5 \pm 4.9$ | $50.6 \pm 4.9$ | $64.5 \pm 4.7$ |
| OCGDT | $\mathbf{51.0 \pm 4.9}$ | $49.8 \pm 4.9$ | / | $51.6 \pm 4.9$ | $\mathbf{53.7 \pm 4.9}$ | $69.1 \pm 4.5$ |
| IQL 800k | $47.5 \pm 4.9$ | $49.0 \pm 4.9$ | $\mathbf{47.9 \pm 4.9}$ | / | / | / |
| IQL 400k | $49.4 \pm 4.9$ | $47.4 \pm 4.9$ | $43.8 \pm 4.9$ | / | / | / |
| IQL 13k | $36.4 \pm 4.7$ | $33.8 \pm 4.7$ | $30.2 \pm 4.5$ | / | / | / |

proves to be the most challenging bot, with OCGDT and ODT achieving similar win rates. The DT-based methods match the results obtained by IQL, requiring less update steps ($\leq 13,000$ vs $\geq 400,000$) to achieve the same results. The models of interest are pitted against each other to further understand the performance differences. Table 2 shows that OCGDT matches win-rates with CGDT, ODT and IQL 800k, the IQL variant trained on 800,000 steps, while getting a positive win-rate against IQL 400k, which is trained for the same amount of wall-clock time, and handily beating IQL 13k, which is trained for the same amount of gradient updates.

## 5.1 ABLATIONS

To evaluate the contribution of each component of the OCGDT model, a series of ablation studies were conducted. The experiments yielded insights into the dynamics of fine-tuning for DT-based agents and identified key sensitivities of the training regimen.

OCGDT *A* uses a trained critic and a policy trained purely on online episodes. As expected, without offline pre-training, the agent performs poorly. A strong policy foundation is essential for reducing the training duration. Our primary investigation focused on the impact of the online fine-tuning phase. We found that extending the fine-tuning duration (OCGDT *B*) degraded performance against more difficult opponents. We hypothesized that this was due to the replay buffer being polluted

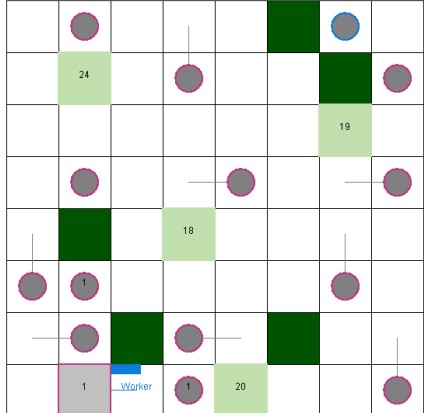

Figure 2: The agent (blue outline) abuses pockets to survive, exploiting the opponent's behaviour.

by sub-optimal trajectories generated during early online exploration. To test this, OCGDT *C* was trained with a larger buffer to preserve the original offline trajectories. While this prevented the degradation seen in OCGDT *B*, it did not yield an improvement over the base model. OCGDT *D* skips the fine-tuning portion. Most surprisingly, the result is on par with the base OCGDT model. With the current dataset, the online fine-tuning mechanism does not provide a significant benefit over the offline model and can be detrimental if not scheduled carefully. The combined results from OCGDT *B*, *C*, and *D* indicate that the agent's self-generated online data may not be of sufficient quality to improve upon the initial policy. This highlights a challenge in ensuring productive exploration for DT-based agents. We also explored the offline training phase itself. In OCGDT *E*, we extended its duration but observed that the agent began to overfit our dataset. This suggests that a larger and more diverse dataset would be required to benefit from a longer offline training schedule.

Finally, we assessed the model's sensitivity to context length. OCGDT *F* and OCGDT *G* are trained with a shorter context $(K = 20)$, with OCGDT *G* trained for twice the duration. These agents match the base agent against bots. This indicates that a long-range context is not a critical requirement for this specific environment configuration, in line with recent findings (Bhargava et al., 2024), suggesting that an agent trained on a longer context is more prone to overfitting with a limited dataset or a Markovian environment. OCGDT *G* empirically demonstrates this compared to OCGDT *E* as the agent's performance did not degrade when trained for longer. Context lengths of 200+ resulted in significantly increased training times and overfitting and are therefore excluded.

## 5.2 LEARNED BEHAVIORS

The agent's general strategy was to overwhelm the opponent with workers. Workers evaded enemy units until friendly reinforcements. In losing positions, the agent placed its final worker in a pocket (see Figure 2) exploiting CoacAI and Mayari's behaviors. The bots preferred building a barracks to create stronger units instead of sacrificing a worker. This tactic allowed the agent to play for a draw.

The agent learned to gather resources from squares that are not directly adjacent to the main base, financing its strategy even after the closest resource node is exhausted. However, this behavior was unreliable beyond four grid squares.

## 6 LIMITATIONS

The ablations expose several limitations which must be taken into consideration in future work. Most notably, the relative ineffectiveness of online fine-tuning compared to purely offline training. Most likely, the agent's policy at the start of the fine-tuning process is not general enough to produce useful trajectories. This pollutes the replay buffer with suboptimal trajectories and hampers the learning process. However, the issue can't be alleviated by simply extending the offline training duration. The training set must be large and diverse enough to support additional offline training without overfitting. Increasing the buffer size to prevent online trajectories from overwriting offline

trajectories too early mitigates the degradation, as this allows the agent to transition more smoothly to online-only trajectories by shifting the trajectory distribution more slowly.

In this work, only two context lengths have been extensively tested: $K = 20$ and $K = 100$, one representing a 'short' context and one representing a 'long' context. The choice of context length is limited by hardware and the quality of the dataset. More GPU memory allows for longer contexts to be attended to. However, a longer context is not strictly better, as it makes the agent more prone to overfitting when compared to shorter contexts, as evidenced by ablations OCGDT $E$ through $G$.

While our experiments are performed on multiple map settings, we only used a map size of $8 \times 8$. Larger map sizes require additional hardware resources, especially to maintain the same context lengths. Thus, this issue is outside the scope of this work.

Given that the µRTS environment does not have a standard offline training set like Minari (Younis et al., 2024), there are no previous benchmarks for state of the art offline RL in the µRTS environment. Therefore, any additional benchmark must be reproduced and optimized for the µRTS environment. IQL was chosen as it was used as one of the benchmarks in both the CGDT (Wang et al., 2024) and ODT(Zheng et al., 2022) papers.

# 7    CONCLUSION

To explore the effectiveness of DT in Gym-µRTS, we reconstructed and applied two DT-based agents: CGDT and ODT. OCGDT is constructed to explore whether the strengths of the two agents can be combined to produce an agent that can both make use of a critic and can perform online fine-tuning. The agents were evaluated against past IEEE CoG µRTS competition winners CoacAI and Mayari, which are rule-based agents. In addition, an IQL agent was reconstructed and modified to fit the state and action space of Gym-µRTS for further evaluation. A dataset was synthesized from 3,000 games between CoacAI and Mayari on procedurally generated maps.

OCGDT matched the performance of both its components against the benchmark bots. In addition, all base DT agents matched the performance of IQL, a state of the art Offline RL agent, against the bots. OCGDT obtained a win rate of $26.2\% \pm 4.3\%$ and $40.1\% \pm 4.8\%$ against CoacAI and Mayari respectively, both of which are past IEEE CoG competition winners. IQL obtained a win rate of $21.5\% \pm 4.0\%$ and $42.6\% \pm 4.8\%$ against CoacAI and Mayari. OCGDT alone managed to obtain a positive win-rate against the three variants of IQL (800k steps, 400k steps, and 13k steps) with win rates of $51.6 \pm 4.9, 53.7\% \pm 4.9$, and $69.1 \pm 4.5$ respectively, over 400 games each.

Training consisted of offline critic (3,000 steps) and policy (5,000 steps) training, and online (5,000 steps) fine-tuning. A dataset of 3,000 trajectories is used, generated by games between CoacAI and Mayari on procedurally generated maps. A single run totaled 4.25 hours of wall-clock time using consumer hardware. DT-based agents are faster to train and require less gradient updates than IQL to match its performance against the bots.

While the ablation with a shorter context length ($K = 20$) showed no performance degradation on 8x8 maps, we hypothesize that longer context lengths will be crucial when tackling larger maps or environments with partial observability. Additional future work includes a richer and larger dataset that can make the offline policy more robust to potential degradation in the fine-tuning stage. Alternatively, established RL techniques with stronger convergence guarantees can be used for fine-tuning in combination with a policy pre-trained on a DT-based architecture.

# 8    REPRODUCIBILITY STATEMENT

The following steps were taken to ensure reproducibility: A link to code and data, including the seeds and hyper-parameter yaml files used to run and train the agents, is provided in Appendix A.1. Relevant architecture details are provided in Section 3 and in Appendices A.2. The hyper-parameter optimization method and the hyper-parameter ranges are provided in Section 4.

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

## A APPENDIX

### A.1 CODE AND DATA

Files for code and data can be found in the following link:

https://osf.io/g4jb6/files/osfstorage?view_only=
bed71727d8c34d8d9f1234be1de1fd26

## A.2 HYPER-PARAMETERS, NETWORKS, AND ADDITIONAL RESULTS

Tables 3 and 4 are the hyper-parameter details used to obtain the final results. The figures show state and action embeddings (Figure 3), OCGDT policy and critic heads (Figures 4 and 5), and IQL policy, Q-, and value networks (Figures 6, 7, and 8). Table 7 shows additional results against weaker rule-based bots.

Table 3: Common OCGDT Hyper-parameters

| Hyper-parameter | Value |
|---|---|
| Optimizer | AdamW |
| Scheduler | OneCycleLR, Sinusoidal |
| Replay buffer size | 3,000 |
| Context length K | 100 |
| Batch size | 32 |
| Initial expectile loss weight $\alpha$ | 0.01 |
| Initial entropy regularization weight $\beta$ | 0.1 |
| Asymmetric expectile loss weight $\tau_c$ | 0.9 |
| Asymmetric expectile loss weight $\tau_p$ | 0.1 |

Table 4: Network-Specific OCGDT Hyper-parameters

| Hyper-parameter | Critic Value | Policy Value |
|---|---|---|
| Dropout | 0.2 | 0.1 |
| Base embed dimensions | 256 | 256 |
| Learning rate | 2e-5 | 1e-4 |
| Weight decay | 2e-4 | 1e-3 |
| Warm-up steps | 1,500 | 2,500 |
| Offline steps | 3,000 | 5,000 |
| Online steps | N/A | 5,000 |
| Transformer heads | 4 | 4 |
| Transformer layers | 2 | 2 |

Table 5: IQL Hyper-parameters

| Hyper-parameter | Value |
|---|---|
| Optimizer | AdamW |
| Scheduler | OneCycleLR, Sinusoidal |
| Batch size | 32 |
| Dropout | 0.1 |
| IQL $\tau$ | 0.1 |
| Soft update factor $\tau$ | 5e-3 |
| Behaviour cloning weight $\beta$ | 4.0 |
| Discount rate $\gamma$ | 0.99 |
| Value function learning rate | 1e-3 |
| Q function learning rate | 5e-4 |
| Actor Learning rate | 5e-4 |
| Reward Scale | 1.0 |
| Steps | 800,000 |
| Warm-up Steps | 360,000 |

Table 6: Parameter counts for ODT, CGDT, OCGDT, and IQL. The DT-based agents share similar parameter counts due to re-using the same architecture. The IQL parameter count excludes the *target* critic network.

| Network | Parameters |
| --- | --- |
| ODT Actor | 17,105,408 |
| CGDT Critic | 4,186,498 |
| CGDT Actor | 17,105,408 |
| CGDT Total | 21,291,906 |
| OCGDT Critic | 4,186,498 |
| OCGDT Actor | 17,105,408 |
| OCGDT Total | 21,291,906 |
| IQL Critic | 3,341,442 |
| IQL Actor | 13,970,304 |
| IQL Value | 1,774,081 |
| IQL Total | 19,085,827 |

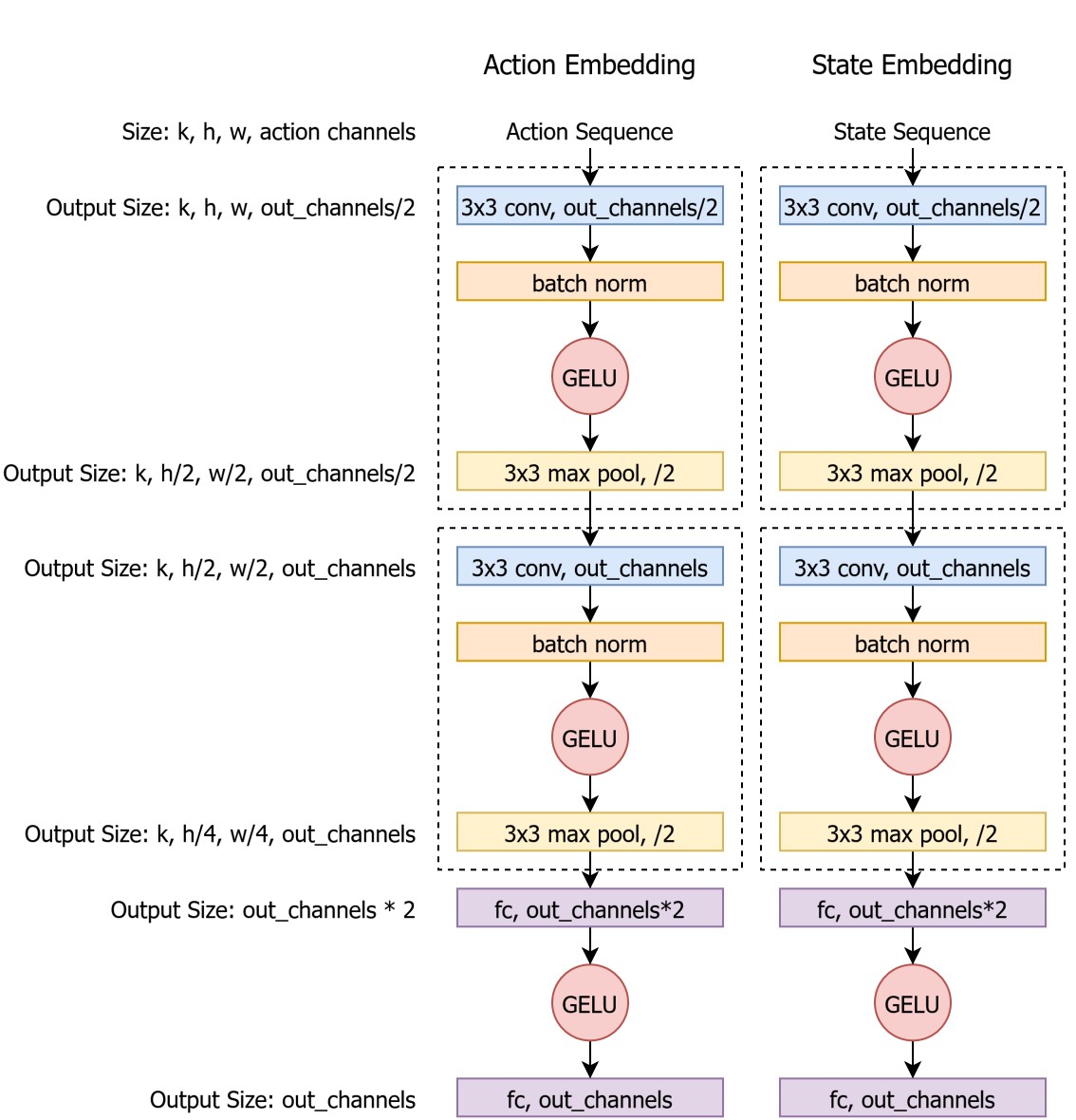

Figure 3: Both action and state embeddings follow the same architecture, the only difference being the number of input channels. CNN outputs are passed through a fully connected layer, a GELU non-linearity, and a final fully connected layer which embeds the inputs into a common shape.

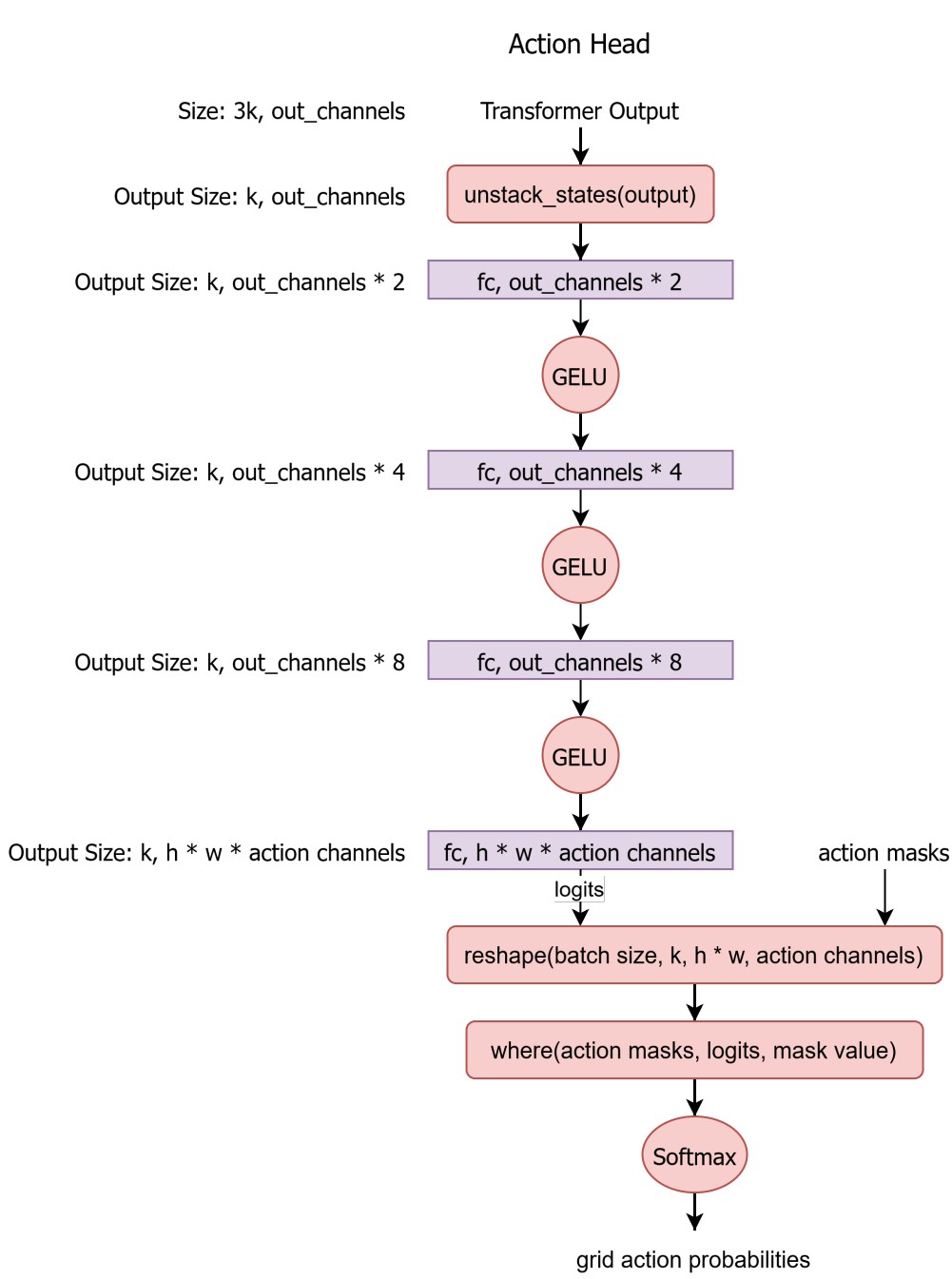

Figure 4: The action head obtains a grid of action probabilities by first passing the transformer output through multiple fully connected layers with GELU activation functions, then reshaping and masking using either an action mask returned from the Gym-μRTS environment or one from the current timestep in the sampled trajectory. Finally, the output channels are split into separate vectors for each action parameter, which is fed to a softmax to obtain likelihoods.

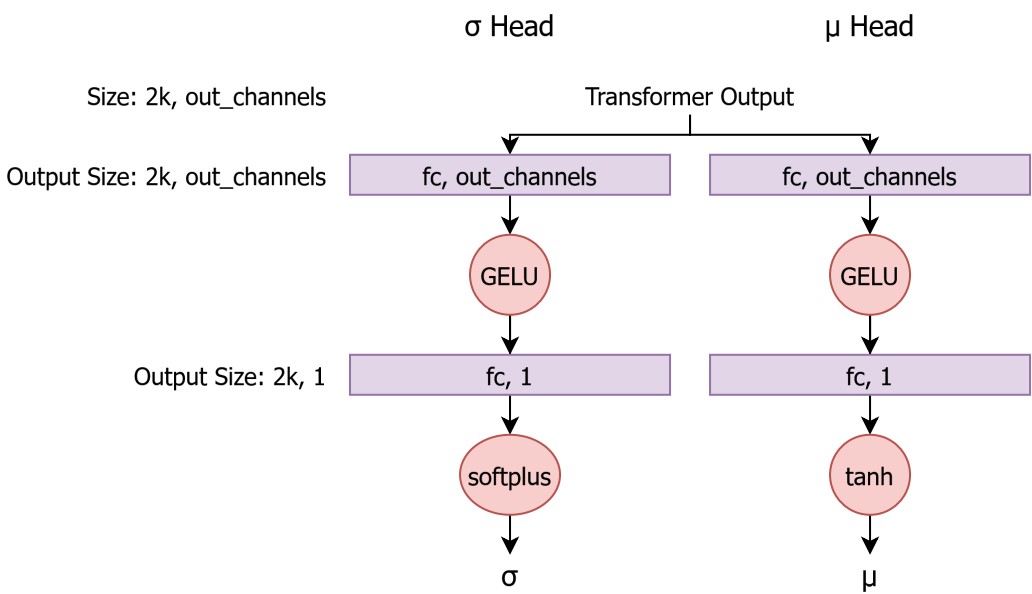

Figure 5: The gaussian parameter heads are the result of a fully connected layer, a GELU activation function, and a final fully connected layer outputting a single value for the head. The value is then mapped to $[0, \infty)$ for $\sigma$ using softplus and to (-1, 1) for $\mu$ using tanh.

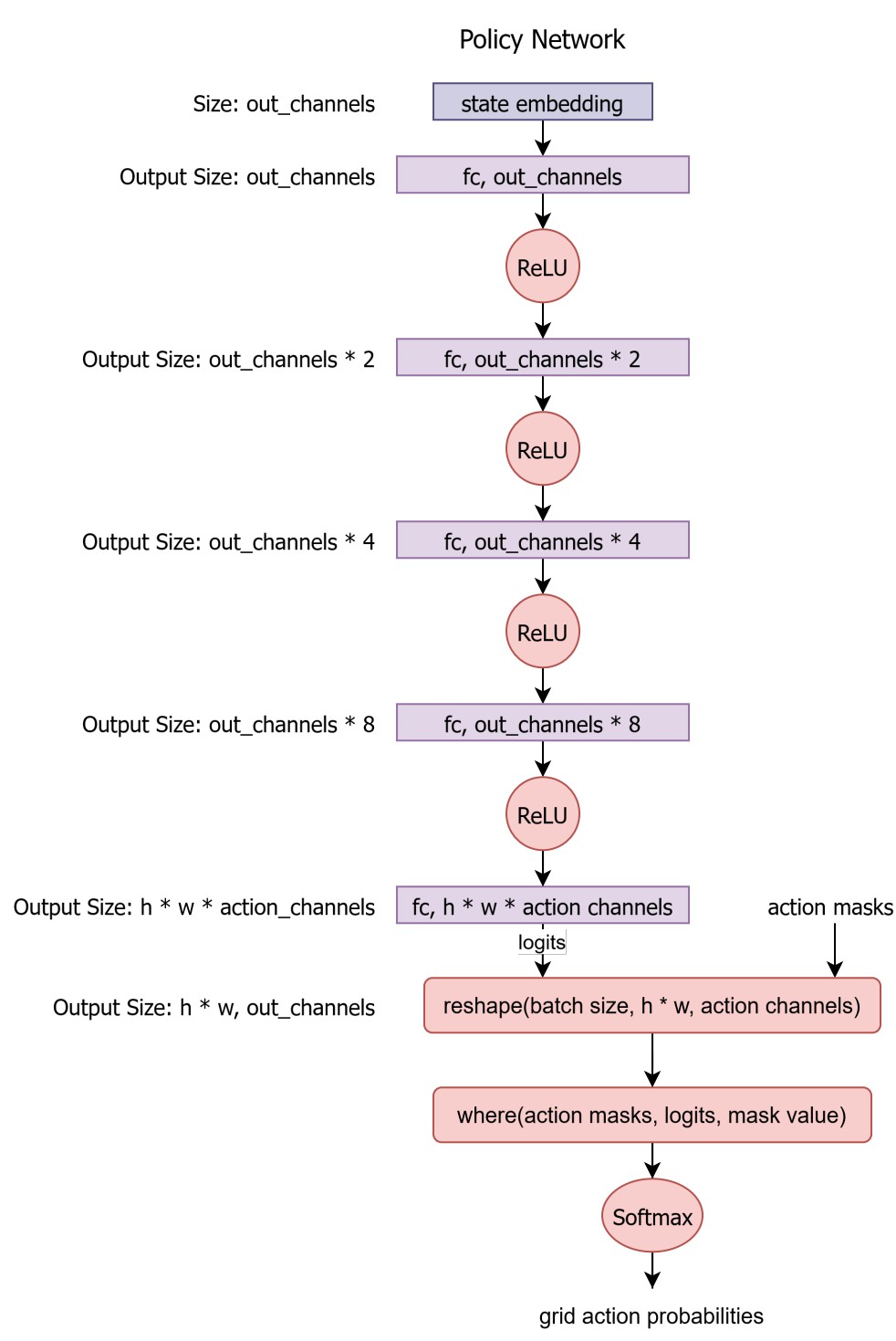

Figure 6: The IQL policy network outputs a grid of action probabilities by first passing the state embedding output through multiple fully connected layers with ReLU activation functions, then reshaping and masking using the action masks in the sampled trajectory. Finally, the output channels are split into separate vectors for each action parameter, which is fed to a softmax to obtain likelihoods.

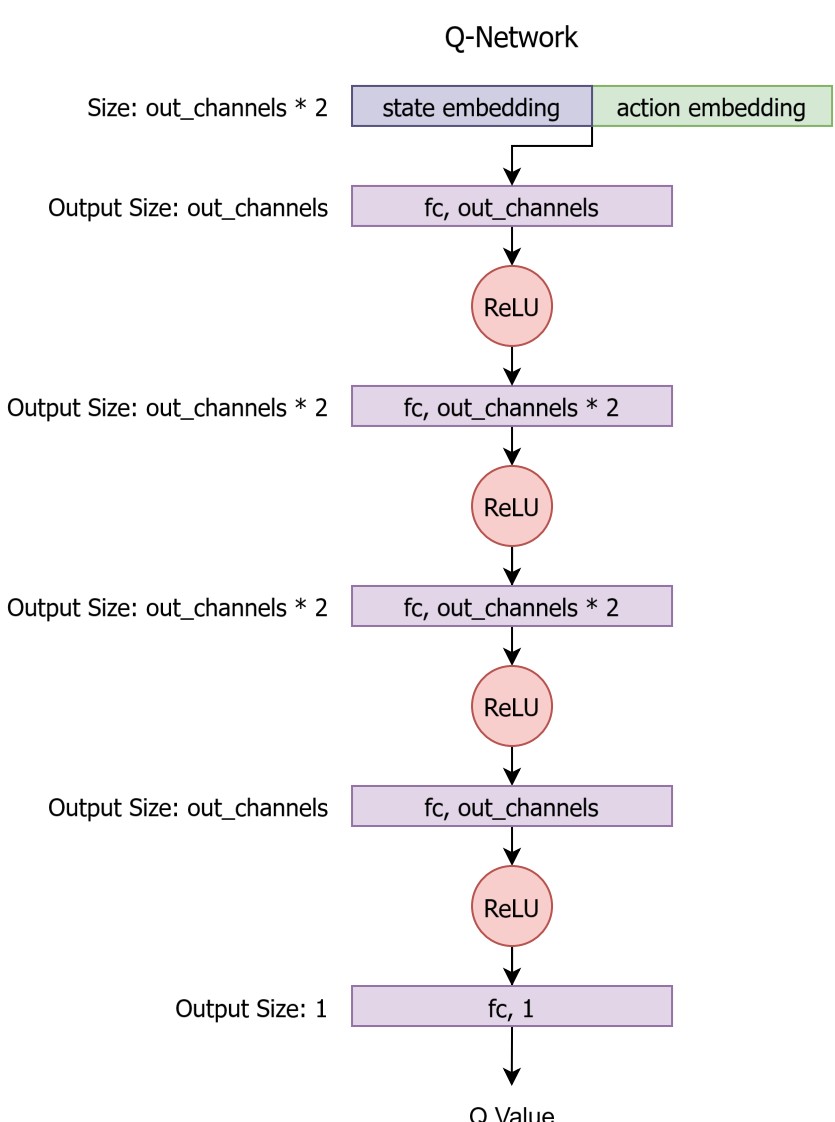

Figure 7: The IQL Q-network outputs a single Q value by taking a concatenated state- and action-embedding and passing them through a series of fully connected layers with ReLU activation functions.

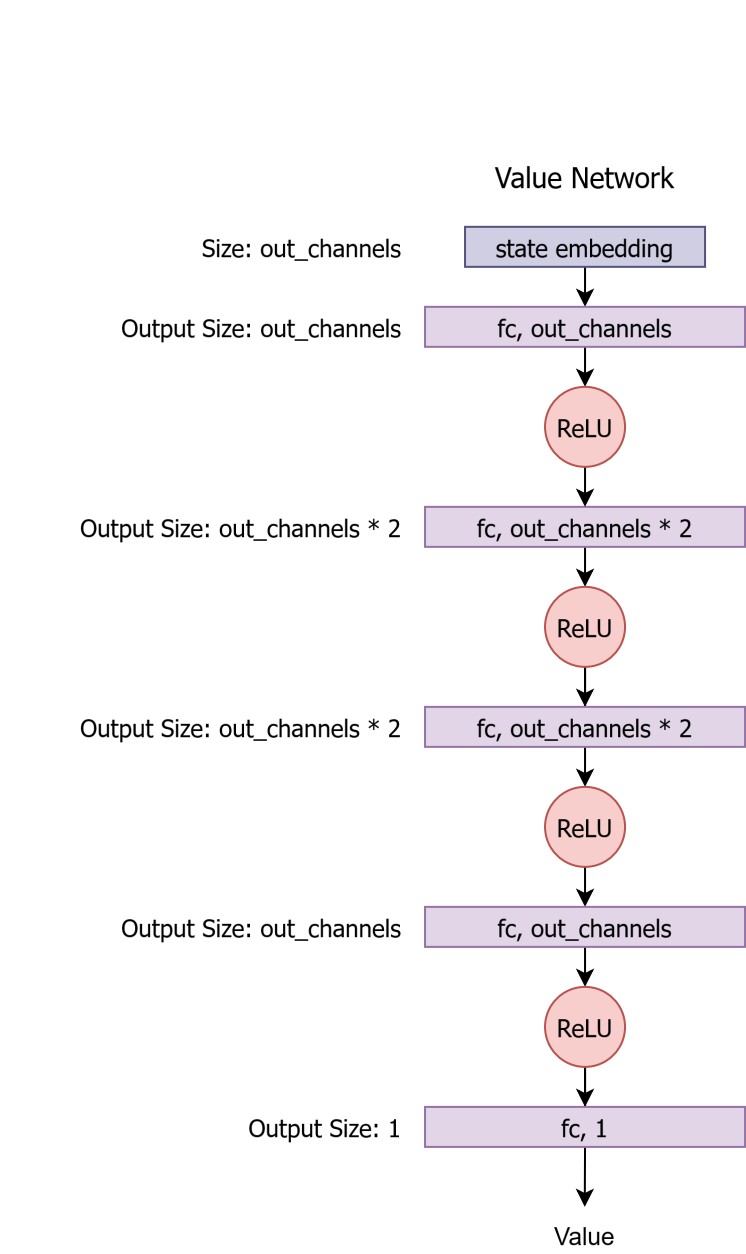

Figure 8: The IQL value nework outputs a single value by taking a state embedding and passing it through a series of fully connected layers with ReLU activation functions.

Table 7: Expanded results from Table 1 including two additional bot agents: workerRushAI and lightRushAI. Agent win rates (%) against benchmark AI bots obtained over 4 seeds, 100 games each, for a total of 400 games on randomly sampled 8x8 maps. OCGDT $A$ is an online-only agent, with a buffer large enough to hold the latest online trajectories. OCGDT $B$ underwent double the fine-tuning steps. OCGDT $C$ underwent double the fine-tuning steps but with an extended buffer size that prevented offline data from being replaced by online data. OCGDT $D$ did not undergo fine-tuning. OCGDT $E$ underwent double the offline training steps (10,000 instead of 5,000). OCGDT $F$ is trained with $K = 20$. OCGDT $G$ is trained with $K = 20$ for twice the duration. Interval represents 95% Wilson score interval. Best performance in bold.

| Method | CoacAI % | Mayari % | workerRushAI % | lightRushAI % | K | Buffer | Offline Steps | Online Steps |
|---|---|---|---|---|---|---|---|---|
| CGDT | $22.3 \pm 4.1$ | $40.8 \pm 4.8$ | $75.0 \pm 4.2$ | $97.8 \pm 1.4$ | 100 | 3,000 | 5,000 | 0 |
| ODT | $25.5 \pm 4.2$ | $\mathbf{46.3 \pm 4.9}$ | $78.2 \pm 4.0$ | $98.0 \pm 1.3$ | 100 | 3,000 | 5,000 | 5,000 |
| OCGDT (Ours) | $\mathbf{26.2 \pm 4.3}$ | $40.1 \pm 4.8$ | $77.2 \pm 4.1$ | $\mathbf{98.5 \pm 1.1}$ | 100 | 3,000 | 5,000 | 5,000 |
| *OCGDT Ablations* | | | | | | | | |
| A (Online Only) | $3.2 \pm 1.7$ | $4.9 \pm 2.1$ | $8.9 \pm 2.8$ | $29.7 \pm 4.5$ | 100 | 4 | 0 | 5,000 |
| B (Double Online) | $15.3 \pm 3.5$ | $29.9 \pm 4.5$ | $68.1 \pm 4.5$ | $97.0 \pm 1.6$ | 100 | 3,000 | 5,000 | 10,000 |
| C (B + Larger Buffer) | $20.0 \pm 3.9$ | $40.8 \pm 4.8$ | $79.0 \pm 4.0$ | $98.3 \pm 1.2$ | 100 | 3,800 | 5,000 | 10,000 |
| D (No Fine-tuning) | $23.0 \pm 4.1$ | $43.3 \pm 4.8$ | $\mathbf{80.5 \pm 3.9}$ | $98.3 \pm 1.2$ | 100 | 3,000 | 5,000 | 0 |
| E (Double Offline) | $16.7 \pm 3.6$ | $35.0 \pm 4.7$ | $70.1 \pm 4.4$ | $96.7 \pm 1.7$ | 100 | 3,000 | 10,000 | 5,000 |
| F (Short Context) | $22.3 \pm 4.1$ | $42.4 \pm 4.8$ | $77.6 \pm 4.1$ | $97.5 \pm 1.4$ | 20 | 3,000 | 5,000 | 5,000 |
| G (F + Double Duration) | $22.6 \pm 4.1$ | $41.1 \pm 4.8$ | $75.4 \pm 4.2$ | $97.7 \pm 1.4$ | 20 | 3,000 | 10,000 | 10,000 |
| *IQL Benchmarks* | | | | | | | | |
| IQL 800k | $21.5 \pm 4.0$ | $42.6 \pm 4.8$ | $75.3 \pm 4.2$ | $97.8 \pm 1.4$ | N/A | 3,000 | 800,000 | 0 |
| IQL 400k | $20.8 \pm 4.0$ | $35.9 \pm 4.6$ | $79.0 \pm 4.0$ | $97.8 \pm 1.4$ | N/A | 3,000 | 400,000 | 0 |
| IQL 13k | $7.9 \pm 2.6$ | $22.1 \pm 4.0$ | $57.1 \pm 4.8$ | $95.9 \pm 1.9$ | N/A | 3,000 | 13,000 | 0 |

