# OpenReview forum: "Controlling a $\mu$RTS agent using Decision Transformers"
_ICLR.cc/2026/Conference — Submitted to ICLR 2026_

### Official Review · Reviewer_c9wo · 2025-10-30

**Soundness:** 2
**Presentation:** 1
**Contribution:** 2
**Rating:** 2
**Confidence:** 3

**Summary:**

This paper implements both Critic-Guided Decision Transformers (CGDT) and Online Decision Transformers (ODT) into the Gym-$\mu$-RTS domain, and also explores the combination of the two methods. The authors build a dataset from games from two previous competition bots  (CoacAI, Mayari), and use this data to train their models, which matches the performance Implicit Q-Learning (IQL).

**Strengths:**

- This paper re-implements two different methods, and combines them together, then applies them to the Gym-$\mu$-RTS domain. Given that the project has been open-sourced, this work may be useful to others.
- The presented algorithm runs on a desktop PC, in a much smaller amount of walltime than many other RL projects. This accessibility and sustainability is something that I believe is often under-valued.

**Weaknesses:**

- Sadly, this paper does not resemble something I would expect to see at a top-tier conference such as ICLR. The writing is quite poor, containing a strangely large number of very short sentences, making it unnatural to read. Furthermore, all of the Figures in this paper are significantly below the typical quality of this venue. I recommend that the authors spend some time reading over previously accepted work and more closely adopting their style.

- The paper only appears to test the proposed algorithm on a single task, with a single set of settings. While $\mu$-RTS is a challenging environment, using a single environment is generally a notable weakness. Testing on different map sizes or settings would have improved the paper.

- The paper has rather limited novelty - it mostly just combines two existing ideas together and adapting them to a new task. While the combination and re-implementation of algorithms can be very useful and sometimes worthy of acceptance, I don’t believe the provided results are groundbreaking enough to justify this.

- The agent’s training data is taken from CoacAI and Mayari, and then also evaluated against these same agents. It is generally poor practice to train and evaluate on the same data/agent.

- The paper does not appear to have a limitations section, which are quite a standard and important section in most papers.

- In Table 1, while I think the ablations were quite interesting, the use of A-G make it quite difficult to read. Please consider using something like OCGDT + Online, and OCGDT + Double Tuning, etc. The caption could still keep the detailed description, but this would make the results easier to digest.

**Questions:**

- In a single paragraph, could you concisely summarize what the novelty of this paper is?
- Can this algorithm be applied to other environments? Could it improve performance?
- Given that this method appears to be very computationally light, perhaps a walltime vs performance graph against prior methods would be a nice way to demonstrate the utility of your method?

---

> ### Author Response · Authors · 2025-11-19
> **c9wo Rebuttal**
>
> Thank you for your review.
>
> Can you please clarify what you mean with regards to poor writing with some examples? Similarly with regards to diagrams, can you please elaborate on their shortcomings?
>
> "The paper only appears to test the proposed algorithm on a single task, with a single set of settings."
>
> While we do not test on different map sizes, we do test on different map settings. We refer the reviewer to section 4.1, lines 303-310 where further details are provided. Notably, we sample from a set of 1,000 procedurally generated maps, each with different starting locations, different resource node locations, and different obstacle locations.
>
> "I don’t believe the provided results are groundbreaking enough to justify this."
>
> With reference to table 2, we emphasise that OCGDT matches IQL's performance in half the wall-clock hours, and outperforms IQL when trained for the same duration. As such, we believe that such a result is noteworthy especially given that it is a novel application of DT methods to this domain. Additionally, it lays the groundwork for future work to further explore the method's strengths and shore up its weaknesses.
>
> "The agent’s training data is taken from CoacAI and Mayari, and then also evaluated against these same agents. It is generally poor practice to train and evaluate on the same data/agent."
>
> While the training data consists of trajectories from Mayari and CoacAI, we vary the map configuration by sampling from a different pool of procedurally generated maps for training, validation and evaluation. The bots behave differently on different maps, resulting in different trajectories.
>
> We include results against the other ML-based agents in Table 2. Additionally, we include further evaluation against weaker bots not used in the training set in the appendix.
>
> The focus of this paper is to tackle the RTS domain. CGDT, ODT, and IQL have already been benchmarked on other domains, so to redo those benchmarks here would be redundant.
>
> "The paper does not appear to have a limitations section [...]"
>
> Noted. A limitations section will be added, summarizing the mentioned limitations in the results and conclusion as well as some additional implementation limitations.
>
> "the use of A-G make it quite difficult to read"
>
> Noted. The labels will be clarified to reflect the ablation.
>
> Q1. With reference to section 1.2 Contributions, this paper is, to our knowledge, the first to implement DT-based methods in $\\mu$RTS. CGDT and ODT are combined to create OCGDT, a new method that uses both a critic and online fine-tuning, matching the performance of IQL when trained for half the wall-clock time while outperforming IQL when trained for the same wall-clock time. OCGDT also outperforms both ODT and CGDT when pitted head-to-head against IQL. Additionally, our training and evaluation sets containing trajectories from CoacAI and Mayari on procedurally generated maps are made publicly available.
>
> Q2. The neural network uses a CNN-based state representation and outputs a grid of actions. This would need to be modified for other environments, but the algorithm itself can remain unchanged. For different settings of the $\\mu$RTS environment, a larger and more diverse training set can certainly improve performance.
>
> Q3. Given that OCGDT is trained in three phases, a running graph of wall-clock time to performance will depend on the length of each of the three phases, making the comparison non-trivial as it would require retraining for each slice of measured time in order to keep the length of each phase proportional. However, a very sparse sample might still be instructive and will be added to the Appendix.

---

> > ### Comment · Reviewer_c9wo · 2025-11-19
> >
> > Thanks for the response, and answering many of my questions.
> >
> > > Can you please clarify what you mean with regards to poor writing with some examples?
> >
> > Take section 3.3 for example. This paragraph is just a list of short statements, with barely any connectives. Each of the statements add almost nothing to the contents of the paper since most of the information should just be in a diagram/table anyway. For example, take these sentences "The first three layers have an output size of 512. The last two layers have an output size of 256 and 1, being the output Q-value. A ReLU activation function is used after each hidden layer." This is just a list of statements, for which the information is already included in diagrams and tables anyway. Just reference a table or diagram unless you have something interesting to say about those design choices. The space in the paper wasted from these lists of statements could be much better used for more evaluation or analysis. As for Figures, take Figure 1 as an example. Lots of text is missing capitalization, the embeddings could say "Action Embedding" and "State Embedding" rather than just "a embedding". For the padding mask, the text looks poor over the top of the graphic, and the braces (denoting k and batch size) shouldn't be touching the graphic. While this may sound like a small aesthetic point to make, all of these combined make the paper feel poorly done.
> >
> > Furthermore, Table 6 doesn't need to be sideways.
> >
> > > I don’t believe the provided results are groundbreaking enough to justify this.
> >
> > I stand by this statement. I think the results are interesting, but not significant enough for an ICLR paper. The improvement adds a substantial amount of complexity for a fairly incremental improvement. While I appreciate work that reduces walltime, this is still not significant enough.
> >
> > > The bots behave differently on different maps, resulting in different trajectories.
> >
> > I still think this is a weak argument. I think your primary evaluation should span many different opponents.
> >
> > > We include results against the other ML-based agents in Table 2
> >
> > The increases in win rate against other agents seems rather marginal compared to existing work.
> >
> > **Question 3 -** Given that a large portion of your contribution is improved walltime, I think it's worth making a Figure which clearly shows the impact.

---

### Official Review · Reviewer_rJBj · 2025-10-31

**Soundness:** 3
**Presentation:** 4
**Contribution:** 2
**Rating:** 2
**Confidence:** 4

**Summary:**

The paper explores an approach to playing real-time strategy games based on decision transformers, and therefore offline RL. The paper leverages two ideas to make DTs more amenable to an RTS setting, which are online DT and critic guided DT. It then combines these approaches to form OCGDT, or online critic guided decision transformers.

The paper converts the RTS setting into an offline RL problem with online fine-tuning by using $\mu$RTS and the associated Gymnasium framework. It collects data using two state-of-the-art, rulebased frameworks for playing RTS games, and both learns from such data and competes with said baselines. The paper also includes an implementation of IQL as an alternative offline RL baseline and compares its contributions with this well-established baseline.

**Strengths:**

The paper is well-written in terms of prose, level of details, and most importantly the clear description of technical details. This is the case throughout, but an example is the description of the architecture; e.g. section 3 does well to describe how the approach "works" and is complimented nicely with figure 1. Several other examples show up in the paper as well.

The idea is clever and the setting is quite interesting given that most offline RL papers are applied to the same benchmarks. There is some technical innovation in converting the problem and getting both the DT-based algorithms and IQL to fit within the $\mu$RTS setup.

**Weaknesses:**

The contribution of OCGDT needs to be made more clear. Is this novel? Non-trivial? The re-implementation of ODT and CGDT on $\mu$RTS problems is interesting in its own right, but again it is unclear if this is the contribution or if it is the creation of a new architecture / algorithm.

In general it is unclear if this approach actually worked. This is perhaps fine given that this is a new foray into this setting from offline RL. But the paper itself mentions that recently ML-based approaches have achieved competitive results in RTS settings. Why, then, is the proposed approach not competitive?

n my opinion, there are several important things to add to the paper (e.g. a more robust IQL description), and hopefully in the main body. Therefore as an editorial suggestion, things like line 334-338, "Training is performed on a Windows 10 machine..." can then be moved to an appendix. These and other similar details are much appreciated and necessary but can likely be moved without degrading the quality of the paper.

**Questions:**

1. The results are somewhat puzzling. My understanding is that the offline data consists of games between CoacAI and Mayari. In the parlance of Offline RL, one might call these "expert" datasets. Why, then, do the methods not achieve parity with either of these baselines? In the case of IQL this is somewhat explanable as it is largely doing imitation learning, and CoacAI/Mayari may be doing things out of distribution at test time. (And if that is the case, is the buffer size appropriate? Should it be larger?) Then, for the online methods, i.e. those of this paper, why don't they perform better?

2. In Table 2, the most interesting thing to me is that CGDT appears to be essentially even wtih IQL (or IQL with sufficient resources). Both of these are offline and therefore, they are imitating each other. Does this suggest that the DT part of the architecture doesn't really matter, and that online finetuning and/or online experience is of first-order importance?

3. Line 423, "This suggests a larger and more diverse dataset...". I agree with this conclusion but I do not think the ablation was necessary to reach it. The fact that IQL has parity with the DT approaches, and that neither are competitive with the expert baselines, suggest that something is amiss with the dataset or perhaps that offline RL is not the correct approach. In standard offline RL datasets, the underlying distributions of the environments are stationary; in the case or RTS, the agent is actually playing a game with the environment, and it (the "environment", which is CoacAI or whatever) changes its distribution according to how OCGDT (or whatever) is behaving. It seems like offline RL will never work for such a case, although the exploration of different and better datasets is encouraged.

4. Line 284: "The actor, the critic, and the value function have separate parameters for state representation". Is this simply saying that there are 3 different neural networks? What is meant by "state representation? The paper could be improved by adding an architectural figure for IQL (probably in the appendix) and making the distinction between IQL details and the various DT details. These are very different things; for example, one doesn't do return-to-go conditioning in standard IQL. Furthermore, is there a transformer in the IQL set up? In other words, is IQL set up with the standard IQL loss functions, the set of neural networks, etc, but that the various neural networks also have transformers? How are they tokenized, and how is this different than DTs (which need return conditioning)? Section 3.3 is probably the least clear part of the paper, and IQL is not really described anyway (while the two variants of DT actually are explained, as well as their combination).

5. The setup to ensure a fair (or "reasonably" fair, or the most fair possible) comparison between IQL and OCGDT is much appreciated. To play devil's advocate, however, this might require a bit more justification. The argument here seems to be about getting an approximately equal number of "experiences" of the data and/or online interactions, and/or gradient steps. This is a solid foundation to start. But each (IQL and OCGDT) have their own hyperparameters at play. So just as a thought experiment, what if Approach A has 100K tunable parameters and Approach B has 100B. In this admittedly extreme example, is it really appropriate to say that having the same "experience" yields a "fair" comparison?

Again, in line 362, DT-based methods require an order of magnitude fewer updates, but are they parameter efficient with respect to IQL? The subsequent text appears to explain this somewhat (i.e. the text about training for wall-clock time and number of gradient steps).

6. Did the authors consider other notions or heuristics for calculating (estimating? Imitating?) the lower entropy bound for $\mu$RTS?

---

> ### Author Response · Authors · 2025-11-19
> **rJBj Rebuttal**
>
> Thank you for your in-depth review.
>
> "The contribution of OCGDT needs to be made more clear."
>
> With regards to contributions, we will clarify section "1.2 Contributions" to further assert the novelty and challenges of this work. Notably, the novel application of DT-based approaches to microRTS, the combination of CGDT and ODT, and the challenges involved, and the trajectory dataset.
>
> "[...] mentions that recently ML-based approaches have achieved competitive results in RTS settings. Why, then, is the proposed approach not competitive?"
>
> We would like to point out the difference in computational resources used by the aforementioned ML-based approach and our approach. RAISocketAI took a total of 70 GPU-days to train the multiple models required for it to become a competition winning agent [1]. For further details on RAISocketAI's training times, see Appendix G - Training Durations in the same paper. In comparison, OCGDT takes 4.25 GPU-hours to train on an RTX 4090.
>
> "[...] there are several important things to add [...]"
>
> Agreed and addressed below with the respective questions.
>
> Q1. It should be noted that Offline RL methods (such as Implicit Q-Learning) and imitation learning methods (such as behaviour cloning) are not equivalent. Offline RL attempts to learn a policy that maximises expected cumulative discounted reward from a fixed set of samples, while imitation learning focuses on matching the actions of an expert without requiring rewards (although they might still be approximated). Inverse Reinforcement Learning methods such as IQ-Learn attempt to recover both a policy and a reward function that justifies that policy. However, Implicit Q-Learning and IQ-Learn are different algorithms from different paradigms.
>
> Additionally, the reviewer states that CoacAI and Mayari produce ‘expert’ datasets. However, this is an assumption. The policies used by CoacAI and Mayari are not guaranteed to be optimal. Further to this, the maps used are procedurally generated and not human-designed, There is no guarantee that CoacAI and Mayari will still produce ‘expert’ trajectories, even if they might have on more traditional human-designed maps.
>
> With regards to fine-tuning, we explore this under-performance in lines 375-376 and lines 414-422 through ablations OCGDT A through D, where we vary the buffer size (4, 3,000, 3,800), and the online step count (0, 5,000, 10,000).
>
> Q2. With regards to imitation learning in IQL, we refer to our answer to Q1.
>
> "Does this suggest that the DT part of the architecture doesn't really matter"
>
> We agree that proper online learning is key. Offline training can be seen as a 'shortcut' alternative to the initial stages of pure reinforcement learning. As such, having a shorter 'shortcut' is useful, and this emphasizes the usefulness of the DT-based methods implemented here given the much shorter wall-clock time required. Note that the 4.25 hour wall-clock time given in the paper is for the full training run (critic + offline + fine-tuning). As such, it takes even less time to reach parity with IQL 800k using offline training alone (OCGDT D, ~2.25 hours vs ~9 hours). The accelerated learning from the DT-based method, while plateauing around the same performance, still matters.
>
> As a side-note, we would like to point out that establishing a parity in performance between DT-based methods and IQL-based methods justifies further exploration of DT-based methods in partially-observable environments, where the context is more likely to make a difference.
>
> Q3. We refer the reviewer to previous applications of Offline RL and Imitation Learning to the RTS domain. [2, 3]
>
> Q4. Yes, the actor, the critic, and the value functions are three neural networks that do not share state representation parameters. We make this distinction with regards to state representation as it is not uncommon to share state representation parameters and simply use different heads for the q/value/actor functions.
>
> The IQL implementation does not use transformers or any form of sequence modeling. It also uses the standard IQL loss functions.
>
> We agree with the suggestion. We will clarify the architectural details in section 3.3 and add an accompanying diagram for IQL's network in the appendix. For further information on IQL, we refer the reader to the original IQL paper, as the focus of this paper is on sequence modeling using DT-based agents.
>
> Q5. Agreed. We will add this metric in the appendix.
>
> Q6. We reused the original heuristic used in ODT with some minor modifications. ODT uses -dim(A). In this paper, we use the same metric, however dim(A) is recalculated at every step to take into account action masking.
>
> [1] - A Competition Winning Deep Reinforcement Learning Agent in microRTS
>
> [2] - AlphaStar Unplugged: Large-Scale Offline Reinforcement Learning (Mathieu M. et al. 2023)
>
> [3] - AlphaStar: Mastering the real-time strategy game StarCraft II (Vinyals, O. et al. 2019)

---

> > ### Comment · Reviewer_rJBj · 2025-11-26
> >
> > I thank the authors for their rebuttal and appreciate the thoroughness of the response.
> >
> > Several concerns have been addressed and, assuming many of these details would be added to a revised manuscript, I believe the paper would be improved. For example, the paper would be improved by making some of the architectural details more clear or prevalent, and/or further discussion on the technical implications and details vis-a-vis _parity_.
> >
> > However, my main concern has not been (and possibly cannot be) addressed. I was asking for clarity on contributions but I am not sure a re-write of the introduction will suffice. I see that the other reviewers have similar concerns. The rebuttals and discourse with the other reviewers echoes many of my own thoughts.
> >
> > The paper is interesting, but at the risk of sounding too informal (and for oversimplifying this paper), it still seems like it falls in the camp of "old solution to a new task" and I am unsure if this is enough.

---

### Official Review · Reviewer_Yhz8 · 2025-10-31

**Soundness:** 2
**Presentation:** 2
**Contribution:** 2
**Rating:** 2
**Confidence:** 4

**Summary:**

This work re-implements two Decision Transformer variants, Online Decision Transformer (ODT) and Critic Guided Decision Transformer (CGDT), along with a widely used offline reinforcement learning method, Implicit Q-Learning (IQL). It further proposes a combined model named Online Critic Guided Decision Transformer (OCGDT) for the Gym-$\mu$RTS environment. Each method is first trained using datasets generated by rule-based $\mu$RTS competition winners, CoacAI and Mayari, and is then finetuned with online interaction. Among the RL approaches, OCGDT achieves the highest win rate against CoacAI, which empirically demonstrates that effective RL optimization in the $\mu$RTS remains a challenging task. Through a range of ablation studies, this paper explores which components of OCGDT are particularly difficult to optimize in the environment.

**Strengths:**

S1. (Clear and reproducible implementation details)
The paper provides detailed descriptions of the RL methods, including their architectures and training procedures.

S2. (Empirical performance of RL methods)
The results show that both DT-based methods and IQL exhibit low win rates when competing against rule-based winners. This finding highlights the difficulty of applying RL methods in the Gym-$\mu$RTS environment.

S3. (Ablation studies)
The ablation study varies several factors, such as buffer size, context window length, and the number of online steps in OCGDT. Through these experiments, it empirically reveals the challenges of online fine-tuning for OCGDT in the Gym-$\mu$RTS. It also emphasizes the importance of appropriately balancing offline data and online samples. However, one of the ablation results remains unclear, as discussed in Weakness W3.

**Weaknesses:**

W1. (Insufficient explanation of RL behaviors)
The paper lacks detailed analysis of how each RL method behaves in the $\mu$RTS. A more thorough explanation would help clarify how these models differ in decision-making.

W2. (Limited analysis of ablation results)
Despite multiple ablation studies (OCGDT A to G) results, their interpretations appear limited. In particular, the difficulties regarding online fine-tuning in OCGDT seem to require additional analysis and discussion.

W3. (Ambiguity in description)
The difference between OCGDT and OCGDT-E in Table 1 is not clearly explained. It would need to specify what distinguishes the two settings and affects their performances.

**Minor**

- typo at line 329; With -> with

**Questions:**

Could you provide further clarification regarding the weaknesses mentioned above, particularly the behavioral explanations of the RL methods and the interpretation of the ablation results?

---

> ### Author Response · Authors · 2025-11-19
> **Yhz8 Rebuttal**
>
> Thank you for your review.
>
> With regards to point 1, we highlight some of the agent's general learned behaviours in section 5.2 (Lines 451 - 458). Is this in-line with the reviewer's expectations? Should there be more focus on analysing learned behaviours such as was done in this section?
>
> With regards to point 2, in lines 375-376, and lines 414-422 we identify the degradation resulting from an extended fine-tuning period, we offer a hypothesis (a replay-buffer polluted with sub-optimal trajectories), and we set up ablations to gain insight (OCGDT C and D). OCGDT C suffers less degradation, while OCGDT D obtains slightly better results than OCGDT C by skipping the fine-tuning portion altogether. We summarise these findings in lines 419-422 and make further reference to this issue in lines 483-485 as future work to further analyse the fine-tuning difficulties and offer a potential avenue for a solution.
>
> With regards to point 3, we describe the difference between OCGDT E and OCGDT in line 383:
> "OCGDT E underwent double the offline training steps (10,000 instead of 5,000)"
>
> However, this issue was also raised by reviewer c9wo, and as such we understand that the letters and the description referencing the letters are not clear enough in distinguishing the different setups. We will update the paper to reflect the configuration differences in the table more clearly.
>
> Thank you for pointing out the typo on line 329.

---

### Official Review · Reviewer_s8iT · 2025-10-31

**Soundness:** 2
**Presentation:** 2
**Contribution:** 1
**Rating:** 2
**Confidence:** 4

**Summary:**

The paper applies Critic-guided Online Decision Transformer to Gym-$\mu$RTS, a long-horizon, stochastic game environment with sparse rewards. OCGDT re-implements and combines ODT and CGDT, two standard return-conditioned sequence modeling methods, to enable offline critic learning, offline policy learning, and online fine-tuning. The authors evaluate against several rule-based bots and run ablations over buffer size, training step, and context length. Empirically, OCGDT matches or exceeds baselines like IQL, CGDT, and ODT.

**Strengths:**

- The paper well summarizes prior work it builds upon.
- The paper includes enough experimental details for reproduction purposes.

**Weaknesses:**

- __The novelty and contribution of this paper are very limited.__ This paper is an application of existing methods to a new task. ODT and CGDT are well-known methods in the literature of RL sequence modelling. OCGDT is merely a combination of both up to some minor changes of the network architecture. And the motivation for combining them does not bring new insights either, as DT's inability of trajectory stitching and suboptimal behavior in the face of environment stochasticity are well-known and ongoing research questions nowadays. OCGDT does not add more algorithmic design for addressing these fundamental issues.

- __The methods involved in the paper are outdated.__ Compared with IQL, ReBRAC [1] is an acknowledged stronger offline RL baseline with the use of actor and critic. For the value-guided DT, QT [2] is one of the current SOTA DT variants. I believe the performance could be stronger when QT and ODT are properly merged.

- __The performance of OCGDT is not appealing.__ In Table 1, OCGDT performs on par with ODT alone for CoacAI, while it performs on par with CGDT and even worse than ODT for Mayari. So, the combination of the two algorithms benefits little. Moreover, IQL is not a proper baseline, since it is not strong enough and it lacks sequence modeling, which is important for this long-horizon, sparse-reward environment.

__References:__

[1] Revisiting the Minimalist Approach to Offline Reinforcement Learning

[2] Q-value Regularized Transformer for Offline Reinforcement Learning

**Questions:**

Please refer to the weaknesses. In addition, could the authors evaluate their method on canonical offline RL benchmarks, e.g., D4RL or Visual D4RL?

---

> ### Author Response · Authors · 2025-11-19
> **s8iT Rebuttal**
>
> Thank you for your review.
>
> "The novelty and contribution of this paper are very limited"
>
> In this paper we are among the first to adapt and apply decision transformers to the $\\mu$RTS domain. Our results empirically show that sequential modeling using decision transformers is competitive with strong traditional offline RL methods like IQL in such a domain. $\\mu$RTS is a complex adversarial environment requiring real-time decision making, long-term dependencies with sparse rewards, and large discrete action- and state-spaces (h x w x 78 and h x w x 29, where h and w = 8 in our case). As such, we believe that such a result is significant enough to be considered a contribution and a stepping stone for future work in this domain.
>
> "The methods involved in the paper are outdated"
>
> Thank you for bringing these papers to our attention.
>
> We acknowledge that ReBRAC outperforms IQL in specific environments. IQL was chosen as it is a strong well-known benchmark with more widespread use as a baseline across the papers referenced by our work. Additionally, we refer the reviewer to Observations 1 and 7 from Tarasov D. et al. (2023) [1], which identify both IQL and ReBRAC as the strongest offline baselines on average (but still not the strongest in every environment, as can be seen by the results).
>
> With regards to QT, this was something that slipped under the radar during initial literature review and we appreciate the reviewer bringing it to our attention. We will strongly consider it in future work.
>
> "The performance of OCGDT is not appealing"
>
> We refer the reviewer to the head-to-head performance of OCGDT vs the other agents (Table 2). Over 400 games on randomly sampled procedurally generated 8x8 maps, our agent matches or outperforms all other agents including in results against IQL. Notably, compared to IQL, OCGDT obtains better performance using half the wall-clock time.
>
> "[...], IQL is not a proper baseline, since it is not strong enough and it lacks sequence modeling, [...]"
>
> With regards to IQL's strength, we refer to our response to the first weakness. With regards to IQL's lack of sequence modeling, we are specifically comparing sequence-modeling methods to a traditional offline RL method to explore their relative strengths. Without a traditional offline RL method as a baseline, the results would only be relevant relative to other sequence modeling methods.
>
> "[...] could the authors evaluate their method on canonical offline RL benchmarks [...]"
>
> Thank you for the suggestion. While it is unlikely that we'll be able to cover the full D4RL suite due to time, we will try to add representative benchmarks to address this.
>
> [1] CORL: Research-oriented Deep Offline Reinforcement Learning Library

---

### Author Response · Authors · 2025-11-19
**Thanks and Update**

Thank you all for the reviews. Newer versions of the paper will be uploaded in the coming days addressing the concerns raised.

---

> ### Author Response · Authors · 2025-11-29
> **Rebuttal Update**
>
> The following changes have been made:
> * Typo fixed (With -> with)
> * Table 1 labels updated (OCGDT A -> A (Online Online), etc)
> * Table 1 format updated to reflect both offline and online steps.
> * Minor changes to 1.2 Contributions
> * Rewording and clarification of 3.3 IQL Architecture with references to new figures.
> * Minor fixes to figures (capital letters, spacing, output sizes)
> * New limitations section: 6. Limitations
> * New parameter count metrics in appendix: Table 6
>
> Unfortunately, it is unlikely that we'll be able to add additional D4RL experiment results and a time-to-performance table of sufficient quality in time.

---

### Meta-Review · Area_Chair_KTam · 2026-01-05

**Summary:**

The paper proposes a method called Online Critic-Guided Decision Transformer (OCGDT) that combines two existing architectures: 1) Online Decision Transformers (ODT) and 2) Critic-Guided Decision Transformers (CGDT). These are pplied to the Gym-muRTS environment.  Authors generate datasets from rule-based systems and train the model in multiple phases: 1) offline critic learning, 2) offline policy learning and 3) online fine-tuning. The key results demonstrate that OCGDT achieves win rates comparable to implicit Q-learning, while requiring significantly less wall-clock training time.

**Reviewer Concerns:**

All reviewers agreed to reject the paper and their concerns were largely unaddressed after the rebuttal period - so there is clear lack of consensus to accept this paper. Reviewers s8iT, rJBj and c9wo mention that the paper represents an application of existing ODT and CGDT methods to a new task without any significant algorithmic innovation. This is a straight forward integration, which is fine if the experimental results were shown on more diverse and realistic environments.

Reviewer s8iT suggested comparisons against stronger baselines like ReBRAC or Q-value regularized transformers. Additionally, while the method matches IQL, it does not significantly outperform the expert baselines it imitates, leading to questions about the efficacy of the approach.

The experimental evaluations are limited to a single environment and relies on training and evaluation against the same set of agents. Multiple reviewers pointed out that the writing quality and clarity needs further work before the paper is ready for acceptance.

**Reviewer Scores:**

s8iT:Kept original score. Felt novelty was limited and baselines were outdated.

Yhz8: Kept original score. Had several concerns regarding behavioral analysis and ablation interpretation.

rJBj: Kept original score. Acknowledged the thorough rebuttal but concluded the work is an "old solution to a new task."

c9wo:Kept original score. Cited limited novelty, single-task evaluation, and presentation issues.

---

### Decision · Program_Chairs · 2026-01-26

Reject